# Thermodynamics of high-pressure ice phases explored with atomistic simulations

Aleks Reinhardt [1], Mandy Bethkenhagen [2], Federica Coppari [3], Marius Millot [3], Sebastien Hamel [3] & Bingqing Cheng[4] ✉

Most experimentally known high-pressure ice phases have a body-centred cubic (bcc) oxygen lattice. Our large-scale molecular-dynamics simulations with a machine-learning potential indicate that, amongst these bcc ice phases, ices VII, VII′ and X are the same thermodynamic phase under different conditions, whereas superionic ice VII″ has a first-order phase boundary with ice VII′. Moreover, at about 300 GPa, the transformation between ice X and the P*bcm* phase has a sharp structural change but no apparent activation barrier, whilst at higher pressures the barrier gradually increases. Our study thus clarifies the phase behaviour of the high-pressure ices and reveals peculiar solid–solid transition mechanisms not known in other systems.

Water ice exhibits remarkable structural diversity: there are currently 20 experimentally confirmed polymorphs with distinct atomic arrangements[1,2]. Intriguingly, at high pressures between 2 GPa to 200 GPa, with the exception of one face-centred cubic (fcc) superionic phase (ice XVIII)[3], all known phases of ices have a body-centred cubic (bcc) lattice of oxygen atoms[4], with subtle differences in the positions and the dynamics of the hydrogen atoms.

Ice VII is the proton-disordered counterpart of the antiferro-electric phase ice VIII, and hydrogen atoms can occupy any of the half-diagonals of the bcc unit cell. Ice VII′ forms upon compression (above ~40 GPa at 300 K) during which hydrogen atoms exhibit substantial translational movements along the bcc half-diagonal, leading to a bimodal distribution in the hydrogen positions between two oxygen atoms. Ice VII″ has a stronger delocalisation of protons compared to VII′, and is the solid phase that coexists with the liquid above ~1000 K. Above ~80 GPa, the bimodal distribution of hydrogen positions becomes unimodal and peaks at the centre: this is ice X. Between ~300 GPa and 700 GPa, density functional theory (DFT) calculations based on the PBE functional predict the P*bcm* phase, with a distorted hexagonal close-packed oxygen lattice, to be stable[5]. It has been proposed that P*bcm* forms due to a dynamic instability in ice X above ~400 GPa[6].

We have only rather limited understanding of the thermodynamic distinctions and the transitions between these high-pressure ice phases. For example, it is not fully resolved whether some of these phase transitions are of first order. The problem is partly due to the limited resolution of the experimental measurements at high pressures, and partly because the system size and timescale accessible to DFT molecular-dynamics (MD) simulations are far from what is required to approach the thermodynamic limit of ice systems. Hernandez and Caracas[7] found a first-order transition between VII′ and VII″ using DFT-MD, based on the sudden changes in the diffusivity, the delocalisation of the H atoms, internal energy and elastic constants. The prediction of this isostructural first-order transition was later corroborated by the experimental evidence of a kink on the melting line $T_m$ at 14.6 GPa, which has been interpreted to correspond to the VII′–VII″ transition line meeting $T_m$[8]. By contrast, for the VII–VII′–X transition sequence, experiments did not find any first-order features in the equations of state[4,9–11]. Similarly, a VII/X equation of state parameterised from DFT-MD simulations suggests no first-order transition occurs, but does not exclude a possible second-order transition[12], and effective potentials for the energy as a function of the proton position between oxygen atoms, constructed from DFT simulations, have been shown to convert continuously from a double well (ice VII) to a single well (ice X)[13]. Some recent work proposed a theoretical framework based on dynamic partition functions to gain insight into the transition between ice VII and its analogues with faster proton dynamics[14]. In DFT simulations, X was observed to transform continuously into P*bcm* upon increased pressure at 0 K without a noticeable barrier starting at about 350 GPa

[1]Yusuf Hamied Department of Chemistry, University of Cambridge, Lensfield Road, Cambridge CB2 1EW, UK. [2]École Normale Supérieure de Lyon, Université Lyon 1, Laboratoire de Géologie de Lyon, CNRS UMR 5276, 69364 Lyon Cedex 07, France. [3]Lawrence Livermore National Laboratory, Livermore, CA 94550, USA. [4]Institute of Science and Technology Austria, Am Campus 1, 3400 Klosterneuburg, Austria. ✉e-mail: bingqing.cheng@ist.ac.at

via a transformation path analogous to that proposed for the bcc–hcp transitions in many metals[5].

From a thermodynamic point of view, it is thus not clear which of these ice phases are actually different phases of water (i.e., separated by first-order phase boundaries, with chemical potentials crossing at coexistence points), rather than the same thermodynamic phase exhibiting different behaviours at different conditions. Determining the nature of these phase boundaries in theoretical calculations entails looking for signs of hysteresis or discontinuities in density or energy during phase transitions, or computing the form of the chemical potentials of the phases. Both approaches require long simulations of large system sizes in order to approach the thermodynamic limit, and consequently standard DFT simulations are generally not feasible because of their severe system size and timescale limitations. Furthermore, the transition pathways amongst the high-pressure ice phases remain largely elusive. Understanding how such transitions occur and in particular the magnitude of the kinetic barrier provides additional insights into the structural relationships between the ice phases, and is also crucial for designing and interpreting high-pressure experiments[3,4,8,15,16].

In this study, we use a first-principles description of high-pressure water based on the Perdew–Burke–Ernzerhof (PBE)[17] approximation to the exchange-correlation functional, as well as a recently constructed machine-learning potential (MLP)[18] fitted to the PBE reference that has been employed to predict the phase behaviour of superionic water. The MLP reproduces accurately the free energies of the superionic water phases of the underlying PBE reference with very small residual error[18]. We first compute the phase diagram of ice at $P < 100$ GPa for the MLP and probe the nature of the phase boundaries between all the ice phases that share the bcc oxygen lattice, and then investigate the X–P$bcm$ solid–solid transition mechanism at higher pressures. Crucially, the combination of first-principles methods, MLP, free-energy estimations and enhanced sampling methods enables us to provide both a thermodynamic picture of the high-pressure ice phases and a detailed mechanistic view of the transitions between them. We not only account for the influence of thermal and quantum fluctuations on the free energies, but also model the dynamic transitions between the ice phases.

## Results
### bcc ices
Between 2 GPa and 200 GPa and below the melting line, except for ice XVIII[3], all the relevant ice phases (VII, VII′, VII″, VIII, X) have a bcc lattice of oxygen atoms and differ by the position and the dynamics of the hydrogen atoms. In this section, we investigate whether there are structural differences between these bcc-based ices or if any differences arise entirely from the dynamics of the H atoms, and whether phase transitions between them are of first order. Unlike previous studies, here we use a thermodynamic approach by focusing on chemical potentials of sufficiently large systems rather than solely relying on the equations of states computed from DFT.

The thermodynamic approach is made feasible by employing the MLP, which has first-principles accuracy but with orders-of-magnitude lower cost. To validate the MLP, we first compare the lattice constants, potential energy and diffusivities computed from the MLP MD simulations with previous first-principles molecular-dynamics (FPMD) simulations[8,19] [Supplementary Fig. 4], which suggests that the MLP is able to capture the dynamic and EOS differences between the different bcc ice phases. Based on the data obtained from brute-force simulation using the MLP, we first make an initial classification of the ice phases in order to compare with the current consensus about the ice phase diagram. In Fig. 1a, we show the low-pressure phase behaviour of the system alongside a line indicating where a discontinuity in density and enthalpy occurs in brute-force simulations along isobars. Beyond ~40 GPa, this line is consistent with the threshold of proton

diffusivity ($D_H > 10^{-8}$ m$^2$ s$^{-1}$) used as a proxy for the superionic transition by Cheng et al.[18]. The background colour in Fig. 1a corresponds to the dominant dynamic phase in brute-force MD simulations started from ice X and equilibrated over long times, using a classification introduced by Zhuang et al.[14,20]: we classify phases according to the frequency of displacements of protons after 0.4 ps by considering where the mode of the displacement distribution is and the skewness of the distribution [Fig. 1b]. This classification enables a qualitative distinction to be made in the behaviour of ice VII and the dynamic nature of the proton diffusion in ice VII′, which we have classed as rotational ('R') and translational ('T') for consistency with ref. 20. We also provide in Fig. 1a an approximate[19] line beyond which the protons exhibit only one peak in the O–H pair correlation function at distances smaller than the nearest O–O distance, which serves to delineate ice X from ice VII.

The qualitative phase behaviour we observe is rather similar to that studied by Zhuang et al.[14,20] and Queyroux et al.[8,19]. The presence of an apparent discontinuity in density and enthalpy in the superionic transition may suggest that this is a first-order transition, although it is often in practice difficult to distinguish between a discontinuous and merely a rapidly changing function. Moreover, for the remaining phases, based on dynamic criteria alone it is not immediately obvious whether they are thermodynamically distinct. To ascertain the nature of the phase transitions in this relatively low-pressure region of the phase diagram, we computed the liquid–solid coexistence curve using thermodynamic integration along isotherms and isobars[21–23] of the appropriate solid and liquid phases starting from an initial coexistence point determined by direct coexistence, and verified by direct-coexistence simulations[24] at several other points. We then determined the VII′–VII″ coexistence line by thermodynamic integration along isobars starting from the liquid–solid coexistence curve. The resulting thermodynamic coexistence curve [Fig. 1a, brown line] largely follows the locus of apparent discontinuities of the enthalpy and density in brute-force simulations; however, despite a relatively low degree of hysteresis, the gradients of the chemical potentials of the two phases of ice are different at either side of the coexistence point [Fig. 1c], confirming that this is indeed a first-order phase transition. The gradient of the $T$–$P$ coexistence curve corresponds to the change in the volume per particle $\Delta v$ and the entropy per particle $\Delta s$ via the Clapeyron equation, $\partial T / \partial P = \Delta v / \Delta s = T \Delta v / \Delta h$, where $\Delta h$ is the enthalpy change per particle. The smaller gradient for the VII′–VII″ transition compared to that of the VII″–liquid and VII′–liquid transitions could arise either from a larger denominator or a smaller numerator, or both. Using the data plotted in Supplementary Fig. 1, we can compute, for example, that at 1400 K, although $\Delta v$ is about 30% larger for the VII″–VII′ transition than for the VII″–liquid transition at their respective coexistence points, $\Delta h$ is even larger, leading to an overall lower $\partial T / \partial P$. It therefore appears that the enthalpy loss, and, in turn, the entropy gain of the highly superionic phase VII″ is the driving force for the VII′–VII″ phase transition, and is the reason that the melting point of the solid is considerably higher than it might otherwise be. However, the liquid–solid coexistence line does not appear to have a significant change in gradient precisely at the triple point; indeed close to the triple point, all three phases have similar densities and enthalpies.

For the remaining ice phases (VII, VII′ (T), VII′ (R), X) shown in Fig. 1a, we observe no clear discontinuities in density or enthalpy, in agreement with experiments[4,9,10] and previous simulation work[12,13]. Moreover, none of these phases appears to be metastable with respect to the others in this class. It is therefore not possible to determine the order of such phase transitions by computing chemical potentials, determining where they cross over and computing their gradients at the binodal point. Instead, we compute the chemical potential by thermodynamic integration along isotherms and isobars across all phases of interest. In principle, thermodynamic integration can only be performed along reversible paths, i.e., without any first-order phrase

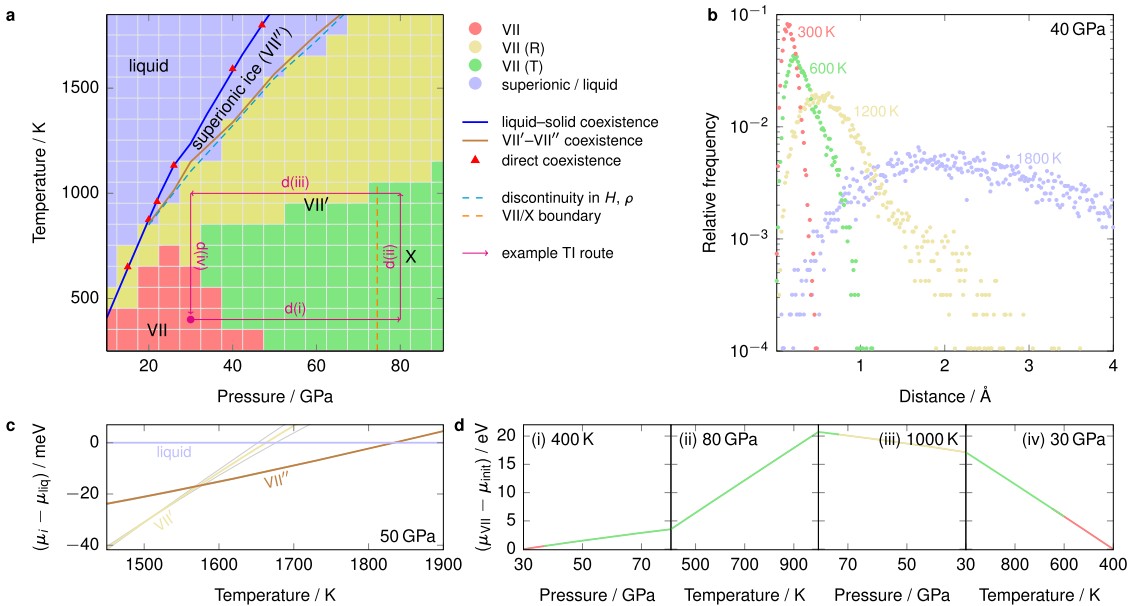

**Fig. 1 | Low-pressure bcc phase behaviour. a** Low-pressure phase diagram. The background colour corresponds to the classification shown in panel **b**. Blue and brown solid coexistence lines were computed from chemical-potential differences using thermodynamic integration (TI). Several points were benchmarked by direct-coexistence simulations at the points indicated. The dashed cyan line corresponds to points where the enthalpy and density are discontinuous along isobars (see Supplementary Fig. 1). The dashed orange line corresponds to the approximate point where the first secondary maximum in the O–H pair correlation function disappears. The thin magenta line gives an example TI route discussed in the main text (see panel **d**). **b** Mean frequency of proton displacement after 0.4 ps for MD simulations started from bcc ice at 40 GPa. Red corresponds to a symmetric unimodal distribution at small displacement ('static ice'). Green corresponds to

distributions with a mode at small displacement and a significant skewness to the right ('ice VII (T)'). Yellow corresponds to a distribution with a mode at medium displacement and a skewness to the right ('ice VII (R)'). Blue corresponds to a unimodal distribution at a large displacement. This classification of phases follows ref. 20. **c** Example discontinuity in the gradients of the chemical potentials for the phase transitions involving the liquid, VII′ and VII″ at 50 GPa. Grey lines are computed from upper and lower limits of the prediction interval at a 95% confidence level for integrands in thermodynamic integration; they are only visible for the VII′ line, since they are narrower than the line width for the remaining phases. **d** Change in chemical potential along the example TI route illustrated in panel **a**. The changeover in colour for ices VII/VII′/X is approximate and based on the classification of panel **b**.

transitions along the way[23]. If we are able to integrate the chemical potential along a 'closed loop' in $T$–$P$ space, we therefore expect that, should any first-order transitions occur along the way, the integral would in principle have a discontinuity and when integrated numerically will result in different values for the chemical potential for the same point when arrived at along different pathways. We have computed the chemical potential along several such pathways, including that indicated by the example route in Fig. 1a. In all cases, we obtain the same chemical potential within numerical error for the same point even when integrated over large regions of $T$–$P$ space and over any combination of the phases shown in yellow, red and green in Fig. 1a. We show an example of the range of chemical potentials along the example TI route in Fig. 1d. Within the numerical accuracy of free-energy calculations, it therefore appears that ices VII, VII′ (T), VII′ (R) and X are not thermodynamically distinct, but merely different manifestations of the same thermodynamic phase under different conditions. The structural difference between ices VII and X in the symmetry of the O–H–O motifs does not appear to give rise to a thermodynamic phase transition. Within numerical error, the density and enthalpy also do not change gradient at the point at which this structural change occurs [Supplementary Fig. 1(ii,vi)] when computed with a finite-difference approach, suggesting that, on the basis of the MLP phase behaviour, there is no evidence that this is even a second-order phase transition in the Ehrenfest sense.

## X–P*bcm* transition

Below the superionic transition temperature, P*bcm* is believed, based on geometry optimisation at zero temperature using DFT, to become stable over X at pressures higher than ~350 GPa[5,6,18]. However, the precise location of the phase boundary at finite temperatures with

the inclusion of nuclear quantum effects (NQEs) is unknown, and the mechanism of this X–P*bcm* transition is not clear. Here we first provide a static-lattice picture of the ice phases at the DFT level, and then exploit the MLP, thermodynamic integration and enhanced sampling to provide both thermodynamic and kinetic descriptions of the X–P*bcm* transition.

First we present the 0 K description for different ice phases predicted by the PBE DFT functional. We considered all the ice structures reported in ref. 18, which were identified by performing ab initio random structure searches (AIRSS)[25]. We computed the 0 K enthalpy at pressures from 100 GPa to 1100 GPa for different ice phases predicted by the PBE DFT functional. The MLP and revPBE0-D3 results are shown in Supplementary Fig. 2; they are qualitatively similar to the PBE results.

In Fig. 2b we show the PBE enthalpy curves for the P4$_2$/*nnm* (ice X(orthorhombic)), P$n\bar{3}m$ (ice X(cubic)), P*bca*, P*bcm*, P2$_1$2$_1$2 and P4$_2$2$_1$2 phases, whose structures are illustrated in Fig. 2a. The P2$_1$2$_1$2 and P4$_2$2$_1$2 structures in Fig. 2b can be considered to be mixed-stacking low-temperature insulating ice structures, and they have excess 0 K enthalpies compared to the P*bcm* phase. Both P*bcm* and P*bca* have a distorted hcp oxygen lattice and differ only in the position of hydrogen atoms at $P > 800$ GPa. In what follows, we focus just on the P*bcm* phase at $100 < P/$GPa$ < 800$, as P*bca*, P2$_1$2$_1$2 and P4$_2$2$_1$2 are either the same or simply have a different stacking sequence in this pressure range.

X(orthorhombic) and X(cubic) both have a body-centred oxygen lattice; however, the cubicity of their unit cells differs above 300 GPa. Our phonon calculations [Supplementary Fig. 5] further show that the X(cubic) structure is at a saddle point of the potential-energy surface of the system above ~400 GPa, as suggested by its imaginary vibrational modes. By contrast, X(orthorhombic) has no imaginary modes

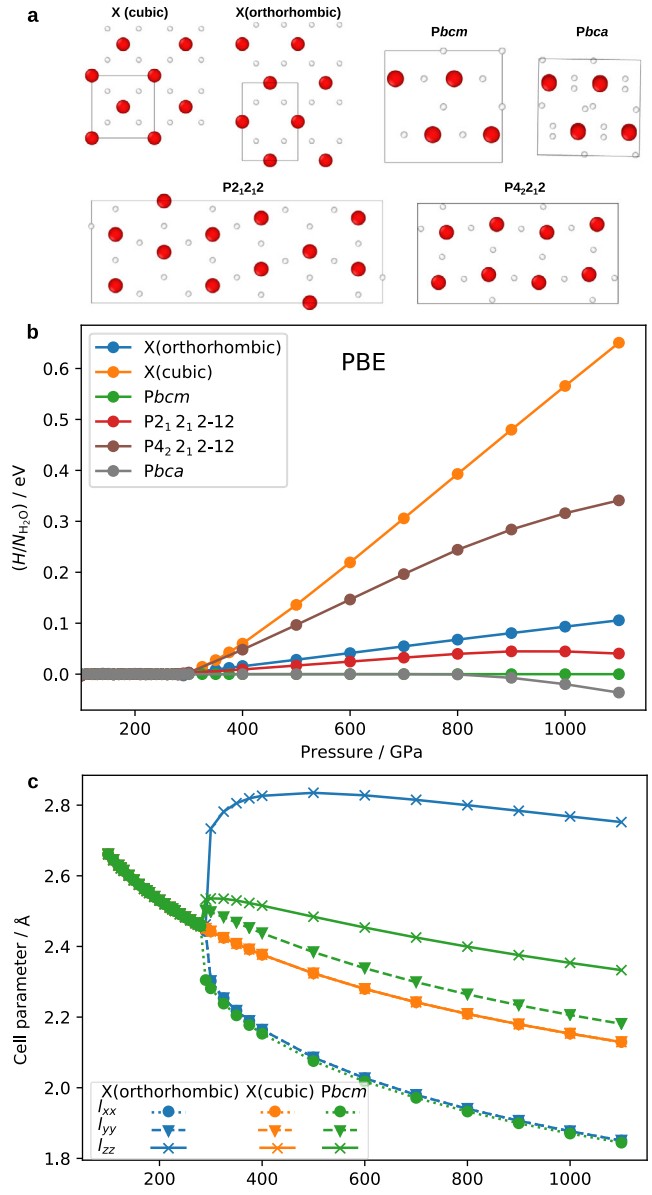

**Fig. 2 | Structures and properties of high-pressure phases at 0 K. a** The structures of the ice phases considered. The simulation supercells used in DFT calculations are indicated using black boxes. **b** 0 K enthalpy curves for different ice phases predicted by the PBE DFT functional. **c** The length of the edge vectors of the simulation supercell ($l_{xx}$, $l_{yy}$ and $l_{zz}$) for X(orthorhombic), X(cubic) and P$bcm$ phases at 0 K. All three cell parameters for X(cubic) are the same, so the three orange curves overlap; two edge lengths for the X(orthorhombic) cell are the same, so two blue curves overlap; and for P$bcm$ there are three different cell parameters. The simulation supercells used for X(cubic) and X(orthorhombic) have 2 water molecules, while the P$bcm$ supercell has 4 molecules (see panel **a**), so the cell parameters for the P$bcm$ supercell in two directions have been divided by $\sqrt{2}$.

in all our phonon calculations performed at pressures between 100 GPa and 600 GPa. Given that the orthorhombic X structure also has a considerably lower enthalpy, we therefore regard it to be the de facto X phase above 300 GPa. In other words, below 300 GPa, ice X has a fully cubic cell (space group P$n\bar{3}m$), and above 300 GPa the cell shape is orthorhombic (space group P4$_2$/$nnm$). We use 'X' to refer to the phase with the cubic–orthorhombic transition at 300 GPa. The breaking of the cubic symmetry of ice X probably arises from the dynamic instability revealed by the imaginary phonons, which was previously reported in ref. 6. A similar breaking of cubic symmetry has

also been observed for other systems[26,27]. Although ref. 6 suggests that the instability leads to the X–P$bcm$ transition, here we suggest that the cubic–orthorhombic transition is an alternative way of gaining dynamical stability.

In the pressure range from 100 GPa to 300 GPa, all the phases considered collapse to the cubic ice-X structure, while the enthalpy difference gradually increases at higher pressures. To probe this collapse further, we show the change in lattice constants as a function of pressure for the X(cubic), X(orthorhombic) and P$bcm$ structures in Fig. 2c. At $P$ < 300 GPa, the cell parameters of the P$bcm$ and the X(orthorhombic) structure indeed correspond to those of X(cubic). Above 300 GPa, the shape of the cell, which is related to the configuration of the ice structures, exhibits a sharp, first-order-like change at the transition. By contrast, the enthalpy (Fig. 2b), the potential energy and the molar volume (Supplementary Fig. 3) all exhibit rather smooth changes across the onset of the bifurcation at 300 GPa. Moreover, although the configurational change is sharp, the transition happens readily during geometry optimisation at 0 K, suggesting a lack of an activation barrier associated with the transitions both between X(orthorhombic) and X(cubic) and between P$bcm$ and X(cubic) at the bifurcation point. To the best of our knowledge, such a bifurcation point has not been observed before in any other systems.

To understand better the nature of the X–P$bcm$ phase transition, we investigated its behaviour at finite temperatures. To this end, we ran well-tempered metadynamics[28] simulations with adaptive bias[29] employing the MLP in the *NPT* ensemble at a temperature of 1000 K, with all the simulation supercell parameters allowed to fluctuate. We employed an orthorhombic supercell of 64 molecules, illustrated in Fig. 3a. The shape and size were selected such that the supercell was commensurate with both the X and the P$bcm$ ice structures, as well as other P$bcm$-like structures with mixed stacking. We used two collective variables (CVs): $CV_1 = [(l_{xx} - l_{yy})^2 + (l_{yy} - l_{zz})^2 + (l_{zz} - l_{xx})^2]^{1/2}$ is based on the anisotropy of the edge vectors of the simulation supercell, whilst $CV_2 = [l_{xy}^2 + l_{xz}^2 + l_{yz}^2]^{1/2}$ measures the tilt of the supercell, whose edge vectors are defined as ($l_{xx}$, $l_{xy}$, $l_{xz}$), ($l_{xy}$, $l_{yy}$, $l_{yz}$) and ($l_{xz}$, $l_{yz}$, $l_{zz}$). It is worth noting that, even if the geometry of the supercell is carefully selected, when probing solid–solid transitions using metadynamics, there can be some hysteresis which affects the computed free-energy profile. Each set of metadynamics simulations lasted 3.5 ns, which is considerably beyond the timescale accessible in DFT simulations, and is only made tractable by using the MLP. To illustrate the validity of the MLP for describing transition processes during metadynamics simulations, we recomputed the PBE DFT energies for 2,500 snapshots collected from metadynamics simulations and demonstrated that the PBE and the MLP energies agree for these snapshots (see Supplementary Fig. 6).

The computed free-energy surfaces (FES) at 200 GPa, 300 GPa, 400 GPa and 600 GPa and a temperature of 1000 K are presented in Fig. 3. At $P$ = 400 GPa, the FES entails five relevant regions: One for ice X, two for P$bcm$, and two (SF1 and SF2) corresponding to structures similar to P$bcm$ but with different stacking sequences. The P$bcm$ phase has two minima on the FES because it can fit into the simulation box in two different ways due to its specific crystal structure. SF1 and SF2 have higher 0 K enthalpies compared to the P$bcm$ structure, and are analogous to the P2$_1$2$_1$2 and P4$_2$2$_1$2 structures illustrated in Fig. 2a.

During the metadynamics run, the system had frequent transitions between the five states. The regions that can be considered as transition states are wide and only have a free-energy excess of ~0.03 eV per formula unit (f.u.). At $P$ = 600 GPa, the FES still has the five regions that correspond to the four distinct ice structures, although they are more separated and the transition states have much higher free energy (~0.18 eV per f.u.). At $P$ = 300 GPa, the five regions are instead connected with no transition barrier. At $P$ = 200 GPa, only one minimum is explored in our metadynamics simulations even when a substantial amount of a biasing potential has been added. This

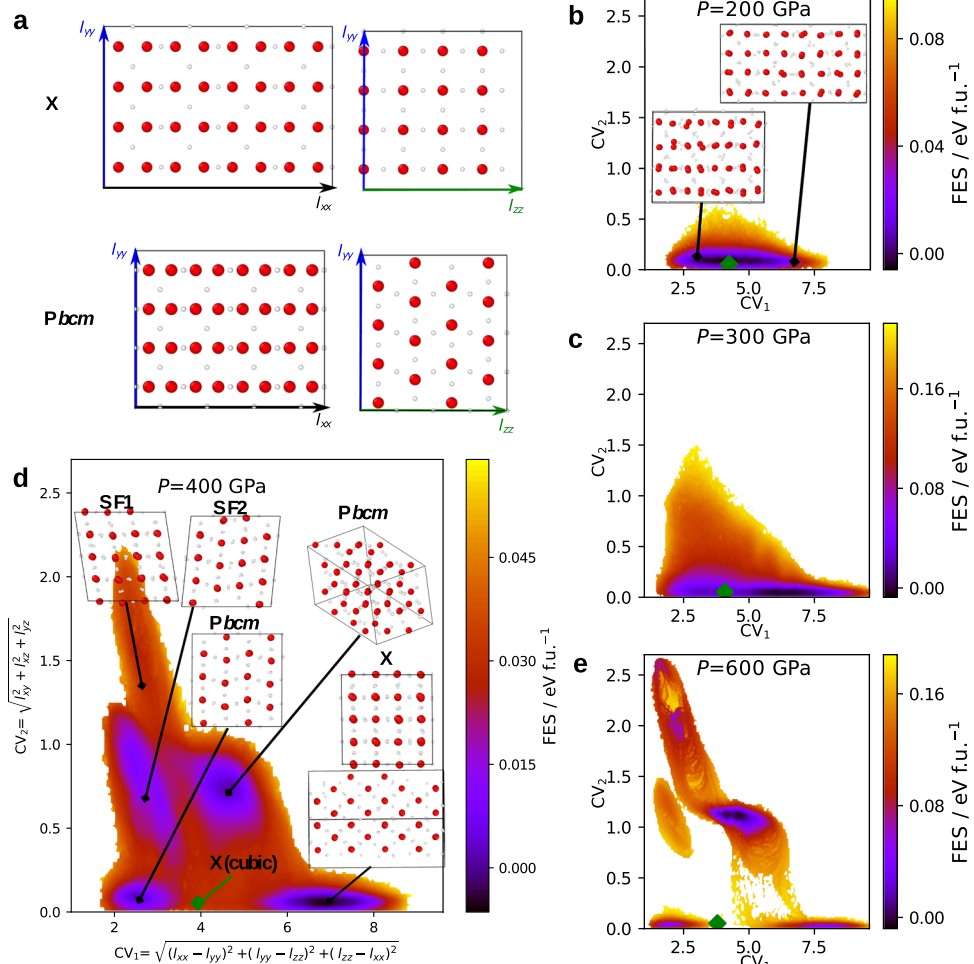

**Fig. 3 | Free energy of the X–P$bcm$ transition. a** Configurations of P$bcm$ and X. The orthorhombic simulation cell contains 64 water molecules and is commensurate with both the X and the P$bcm$ structures. Free-energy surfaces (FES) at **b** 200 GPa, **c** 300 GPa, **d** 400 GPa and **e** 600 GPa, all computed at 1000 K, as a function of two collective variables: $CV_1$ measures the anistropy of the box and $CV_2$ measures the tilt of the box. Representative snapshots of the ice structures collected during the metadynamics simulations are shown. SF1 and SF2 are structures that are similar to P$bcm$ but with different stacking sequences. The green diamond symbols indicate the location of X(cubic) on the FES.

minimum corresponds to the X phase, and the main variance within the minimum is associated with the elongation of the supercell. In addition, we use green diamond symbols to indicate the location of X(cubic) in the space of the CVs in Fig. 3. It can be seen that at 200 GPa, X(cubic) is in a minimum on the free-energy surface, and at higher pressures it becomes a saddle point.

The FES in Fig. 3 again confirms the bifurcation transition revealed in the 0 K enthalpy curves in Fig. 2: at high pressures ($P \geq 300$ GPa), there are distinct phases including P$bcm$, X and the phases with stacking faults, while at low pressures ($P \leq 300$ GPa), all the phases collapse into X. These solid–solid transitions are more facile at lower pressures, as suggested by the lack of activation barriers at 300 GPa and the lower barriers at 400 GPa compared to 600 GPa. The transition pathway from X to P$bcm$ involves a highly collective shuffling of the {100} planes in X, similar to the bcc–hcp transitions in many metals[5,26], and therefore does not entail a nucleation step. We also performed metadynamics simulations using system sizes 8 times as large (see Supplementary Fig. 7 for snapshots of atomic positions); the transition mechanism of the diffusionless plane shuffling is identical in these larger systems. Finally, to validate that the MLP can correctly capture the dynamics of this transition process, we have run both brute-force MLP and PBE MD simulations at 1000 K and 400 GPa for 12.5 ps. In both cases we observed the facile, diffusionless transition from X(cubic) to P$bcm$ on the same timescale.

Both the static-lattice enthalpy curves and the results from finite-temperature metadynamics simulations suggest that P$bcm$ and X are distinct phases above ~300 GPa, but collapse into a single phase at lower pressures. Where they are two separate phases, one can ask what their relative thermodynamic stabilities are and how the phase boundary depends on temperature. Moreover, the 0 K enthalpy curves only provide a partial description of the chemical potentials: although P$bcm$ has a lower enthalpy at the static-lattice level, the excess enthalpy of the X phase is rather small, and thermal and quantum fluctuations can have a strong influence on the thermodynamic free energies for water systems[21,30].

Although X and P$bcm$ appear to be identical below 300 GPa, during the free-energy calculations using thermodynamic integration, we choose to be agnostic about this fact and instead always treat P$bcm$ and X separately at all pressures. Generally speaking, when computing the classical Gibbs energy of a system, TI can be performed between the reference crystal and the classical physical system at a certain $P$-$T$ point (the so-called $\lambda$-TI, or hamiltonian TI[31]), or along isotherms and isobars to calculate the chemical-potential differences at other pressure and temperature conditions. It is worth noting that, as the X(cubic)–X(orthorhombic) and X(cubic)–P$bcm$ transitions at about ~300 GPa are peculiar and accompanied by sharp jumps in the lattice parameter (Fig. 2c), although the enthalpy changes are smooth, it is not completely clear whether the TI method

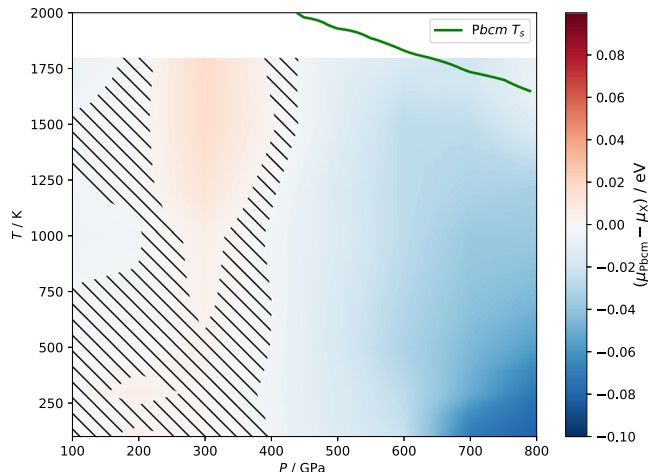

**Fig. 4 | High-pressure phase diagram.** The chemical-potential difference $\mu_{Pbcm}$ − $\mu_X$ per formula unit (f.u.) between P$bcm$ and X as a function of temperature and pressure. The statistical error in the chemical-potential difference results in the uncertainty of the coexistence line, which is indicated by the hatched area. The green curve shows the insulating–superionic transition temperature $T_s$ defined by a threshold of proton diffusivity ($D_H > 10^{-8}$ m$^2$ s$^{-1}$) for the P$bcm$ phase.

still applies along the pressure across the transition boundary. To clarify this ambiguity, we performed multiple TI calculations using the MLP starting from the reference crystals at different pressures from 200 GPa to 700 GPa in 100 GPa steps, and at several temperatures (100 K, 300 K, 600 K, 1000 K). As shown in Supplementary Fig. 8, these different TI routes provide consistent estimations of the classical chemical potentials for both phases employing the MLP, which not only verifies that these values are statistically converged, but also suggests that TI can be used to cross the transition boundary at ~300 GPa. This again highlights the peculiarity of the bifurcation transition, which has continuous enthalpy and free energy, but sharp changes in structures.

To determine the chemical potentials of X and P$bcm$ at the PBE level that include NQEs, we add together the classical chemical potential at the MLP level, a correction term in the free energies that removes the small residual errors in the MLP[30,32], as well as a term that accounts for the influence of NQEs. The resulting phase diagram is shown in Fig. 4. The chemical-potential difference between the two phases is generally very small across the whole temperature and pressure range that we consider. There is essentially no difference between the chemical potentials of X and P$bcm$ at $P < 400$ GPa. Indeed, the structures of X and P$bcm$ at low pressures are the same in that region. The small difference at higher temperatures and low pressures is due to the numerical error in the integration. However, at higher pressures, P$bcm$ becomes more stable. Within statistical uncertainty, the finite-temperature phase diagram is thus consistent with the 0 K enthalpy results of Fig. 2a. There is a weak temperature dependence of the chemical potentials, with higher $T$ slightly favouring the stability of X.

## Discussion

In this study, we combined a PBE DFT description and a machine-learning potential to investigate the nature of the phase transitions between high-pressure phases of insulating ice. Crucially, our approach probes the thermodynamic behaviours of the ice phases. By accurately computing the chemical-potential differences between the phases, we can classify whether these phases are thermo-dyanmically distinct, and determine the orders of the phase transitions between them.

The superionic transition in bcc ice phases has been studied in both theoretical and experimental work[7,8,15,18,19,33–37], although the range of

conditions at which superionic phases are observed is subject to considerable variation in the literature. Although there is a degree of superionicity in ice VII′, the hydrogen diffusivity increases significantly in ice VII″[19]. In experiment, several signatures of a first-order-like phase transition have been reported, particularly in the context of a discontinuity in the gradient of the $P$–$T$ coexistence curve between the liquid and a solid phase, which has sometimes been interpreted as a triple point. This has been reported under a range of different conditions, from 35 GPa and 1040 K[35] to 43 GPa and ~1600 K[36] to 14.6 GPa and 850 K[8], and the solid phases that apparently coexist with the liquid have occasionally been reported to be different, e.g., VII and X in ref. 36. We have obtained a triple point between the liquid, VII′ and VII″ at ~20 GPa and 875 K, close to the value reported by Queyroux et al.[8]; moreover, we have shown that VII′ and VII″ are thermodynamically distinct phases, as suggested by the apparent (though often small) discontinuities of the enthalpy and density in simulations using system sizes large enough to investigate thermodynamic behaviour. We have explicitly computed their chemical potentials to confirm that the VII′–VII″ phase transition is of first order, and we determined the associated thermodynamic coexistence curve between them. We speculate that one possible reason for the various different reported triple points is that in our simulations, it does not appear that the solid–liquid coexistence curve changes gradient significantly precisely at the triple point; there is a more gradual change of gradient as higher pressures and temperatures are reached, as the supercritical ice phase becomes progressively more different from its low-temperature less-disordered analogue.

In X-ray diffraction experiments, no clear distinction has been seen between the other bcc ice phases, including VII and X[4,9,10]. We confirm here that ices VII, VII′ (T), VII′ (R) and X show no clear discontinuities in thermodynamic quantities including density and enthalpy, and in turn the chemical potentials. The only differences between them come from the positions and the dynamics of the hydrogen atoms. We thus regard these phases to be merely different manifestations of the same thermodynamic phase under different conditions.

For the insulating ice phases that have been theoretically predicted for the pressure range between 100 GPa and 800 GPa, we find that all of them (X(orthorhombic), X(cubic), P$bca$, P$bcm$, P$2_12_12$ and P$4_22_12$) collapse into ice X below 300 GPa. This means that there is no distinction between them at pressures from 100 GPa to 300 GPa. Moreover, the transition barriers between X, P$bcm$ and the structures with stacking faults increase as the pressure increases. Above 300 GPa, X and P$bcm$ are thermodynamically distinct phases, and we have computed their relative chemical potentials, fully accounting for thermal and nuclear fluctuations. P$bcm$ has not yet been experimentally observed, despite past and on-going dynamical compression experiments. Our computed phase diagram in Fig. 4 along with the peculiar solid–solid mechanism we have identified provide the following predictions to guide future experiments: (i) The phase boundary between P$bcm$ and X is largely independent of temperature. By contrast, as suggested in ref. 18, the P$bcm$ phase becomes superionic and its distorted hexagonal close-packed (hcp) oxygen lattice becomes fully hcp at the superionic transition temperature $T_s$. This $T_s$, which is marked by the green curve in Fig. 4, is ~1800 K between 300 GPa and 800 GPa. Above this temperature, the superionic fcc phase (ice XVIII[3]) and the superionic hcp phase were predicted to have similar chemical potentials and were both regarded as thermodynamically stable between about 100 GPa to 800 GPa[18]. This means, if feasible, keeping the temperature low (at least below the superionic transition temperature) is essential for making the P$bcm$ phase in compression experiments. (ii) Between 300 GPa and 800 GPa, the chemical-potential difference $\mu_X - \mu_{Pbcm}$ increases with increasing pressure, although the magnitude of the chemical-potential difference remains rather small even at 800 GPa. This means the equilibrium driving force favouring the thermodynamic stability of the P$bcm$ phase

is greater at higher pressures. On the other hand, the activation barrier for the X–P$bcm$ transition also becomes larger, meaning that the kinetic transition rate is smaller at higher pressures. This implies that, in compression experiments that aim to make the P$bcm$ phase, instead of targeting the highest pressure achievable, it may be advantageous to aim for pressures that are moderately above the X–P$bcm$ transition line in order to utilise the lower kinetic barrier. (iii) If the P$bcm$ phase, or a mixed-stacking phase, is indeed made during compression experiments, it will not remain stable and will instead reversibly transform into X whenever the external pressure drops to below 300 GPa.

In summary, our study provides a thermodynamic view of the high-pressure phase diagram of ice below the superionic transition line. Our simulation results suggest that the various bcc ice phases, including ice VII, ice VII′ (T/R) and ice X, are indeed the same thermodynamic phase that behaves differently at different conditions. Our simulations also reveal the peculiar bifurcation and transition pathway between X and P$bcm$, which sheds light on the viability of the experimental synthesis of the as yet experimentally undetected, but long predicted, P$bcm$ phase. Beyond the water ice system, our methodological framework, which combines first-principles methods, machine-learning potentials, free-energy methods and enhanced sampling, can be applied to study the phase diagrams and phase transitions of other polymorphic systems, such as high-pressure solid hydrogen[38], perovskites[39] and two-dimensional materials[40].

Experimental investigations on the high-temperature and high-pressure water ices are notoriously difficult, but recent technical breakthroughs could open new opportunities to test the predictions from this work in the future. For example, neutron scattering was used to investigate the atomic structure of ice VII at room temperature up to ~100 GPa and revealed subtle pressure-induced changes of the proton disorder[41]. Pulsed internal heating and alternating-current calorimetry in the diamond anvil cell (DAC) is a promising technique for investigating the nature of phase transitions experimentally[42,43]. A combination of resistive heating, near- and mid-infrared laser heating in the DAC to study the structure of dense ice at high temperature with X-ray diffraction has provided evidence for phase transformations consistent with two superionic ices having either a bcc or a fcc oxygen sublattice. $CO_2$ laser heating in the DAC has also recently been shown to release deviatoric stresses and enable collection of high-quality powder X-ray diffraction and Raman scattering[44]. Finally, laser-driven dynamic compression has enabled the investigation of the thermodynamics and atomic structure to several hundred GPa and revealed evidence for superionic ice XVIII[3,15]. With the rapid evolution and improvement of experimental methods to investigate material properties at extreme conditions, we expect that this work will provide insights into the interpretation of experimental data as well as input for new experiments.

The precise nature of phase transformations in dense ices at high temperatures could also affect our understanding of large icy worlds in our solar system such as Ganymede, Titan and Callisto[45] as well as more distant water-rich exoplanets[46–49], as the presence (or absence) of discontinuous phase transitions can induce (or suppress) thermal boundaries, which would affect mass and heat flows and have a significant role in shaping the internal structure and evolution of these bodies.

## Methods

### DFT calculations
The DFT calculations were carried out with VASP 5.4 using the hard PAW pseudopotentials and an energy cutoff of 1000 eV. The $k$ points were chosen automatically using a Γ-point-centred grid with a resolution of 0.3 Å$^{-1}$. We considered nine structures with the space groups P4$_2$/$nnm$ (ice X(orthorhombic)), P3$_2$21, P4$_3$22, P2$_1$2$_1$2, P4$_2$2$_1$2, P$bca$, P$bcm$ and Pn$\bar{3}$m (ice X(cubic)), which were taken from a previous

study[18]. That study predicted the structures with AIRSS[25] using CASTEP to perform the DFT calculations and applying various exchange-correlation functionals. Their identified PBE structures were reoptimised in this study with VASP at constant pressure to remove any residual forces. We considered a pressure range between 100 GPa and 1100 GPa and checked that the initial space group for each considered structure was maintained after reoptimisation. The energies were typically converged to 10$^{-8}$ eV and the forces to 10$^{-6}$ eV Å$^{-1}$. Subsequently, the structures were optimised similarly with the revPBE0-D3 exchange-correlation functional to investigate the influence of a hybrid functional on the structural stability. Finally, we calculated the phonons for the P4$_2$/$nnm$, P$bcm$ and Pn$\bar{3}$m structures at the PBE level to identify possible pressure ranges of dynamical instability. Those calculations were performed with Phonopy[50] on 2 × 2 × 2 supercells in 25 GPa steps between 100 GPa and 400 GPa. The phonon densities of states were evaluated with at least 71 mesh $q$ points in each direction.

### MLP MD simulation details
We performed *NPT* MD simulations for bcc ice systems with LAMMPS[51] with an MLP implementation[52]. Bulk phases were simulated in the *NPT* ensemble using the Nosé–Hoover isotropic barostat[53] and a timestep of 0.5 fs, with typical MD simulation runs of between 20 ps and 60 ps at each set of conditions considered. Bulk-simulation system sizes ranged between 128 and 2048 water molecules to check for finite-size effects; for direct-coexistence simulations, we used system sizes of between 2916 and 3453 molecules. We have repeated simulations of ice VII with several different choices of proton disorder. In simulations along isotherms and isobars, we gradually changed the pressure and the temperature in steps of between 0.3 GPa and 4 GPa and between 5 K and 60 K, respectively, with step sizes chosen to be smaller closer to the limits of metastability of each phase. Since each simulation was started from an equilibrated system very close in pressure and temperature, the initial equilibration prior to data collection could be brief.

To compute chemical potentials and in turn the phase diagram, we started by considering a large direct-coexistence simulation[24] of liquid water in contact with ice and determined the coexistence temperature at pressures of 20 GPa and 26 GPa, and the coexistence pressure at 1800 K. These points match very well the coexistence curve reported in ref. 18. At these coexistence points, the liquid and the appropriate solid phases have the same chemical potentials. Finally, we computed thermal averages of system properties along isotherms and isobars in MD simulations, and numerically integrated[23] the Gibbs–Duhem relation, d$\mu = v$ d$P - s$ d$T$, at constant temperature to give the chemical potential as a function of pressure $\mu(P) = \mu(P_1) + \int_{P_1}^{P} v(P')$ d$P'$, where $v$ is the volume per formula unit, and the Gibbs–Helmholtz equation at constant pressure to give the chemical potential as a function of temperature, $\beta\mu(T) = \beta_1\mu(T_1) - \int_{T_1}^{T} (H/Nk_B(T')^2)$ d$T'$, where $\beta = 1/k_B T$ and $H$ is the enthalpy. This enthalpy includes kinetic energy contributions, i.e., the de Broglie thermal wavelength's temperature dependence is accounted for in the chemical potential[54]. In each case, we fitted the integrands to a polynomial to be able to extrapolate them slightly beyond the regions of metastability. We confirmed that starting from any of the direct-coexistence starting points, we can recover the other two, and others, by this thermodynamic integration route. To find other coexistence points, we computed the chemical potentials for all phases that are metastable at a given temperature and pressure, and determined numerically the point at which the chemical potentials of different phases cross over.

In free-energy calculations, it is often difficult to determine the errors in chemical potentials systematically[23], and consistency checks are often performed instead[55], such as using direct-coexistence simulations and computing the chemical potential at a given point arrived at via several independent pathways. In addition to these, we have

estimated numerical errors in thermodynamic integration by computing mean prediction bands for polynomial fits to the data points at a 95% confidence level and propagating the upper and lower band through the subsequent integration. In the vast majority of cases, these confidence bands are narrower than the line widths, but can be larger whenever extrapolations are needed, as shown in Fig. 1c; however, it should be borne in mind that systematic errors such as the choice of initial integration point from direct-coexistence simulations, are likely to be larger than numerical errors, and consistency checks are therefore still essential to perform.

### The chemical potentials of X and P*bcm* phases

We first computed the classical Gibbs energy for both P*bcm* and X using a system size of 512 formula units. We performed geometry optimisation for each structure at different pressures. Where a local minimum was found, the hessian matrices of the structures were computed using the programme i-PI[56] with a finite-difference method. The hessian matrix enables us to construct a reference harmonic crystal in which the forces between the atoms are defined by the phonon modes. After that, we ran hamiltonian TI between the reference crystal and the classical physical system. We used TI along isotherms and isobars to calculate the chemical-potential differences at other pressure and temperature conditions. To validate that we can reach consistent estimates of the chemical potential using different TI pathways, we therefore performed multiple TI calculations: (i) $\lambda$-TI at different temperatures (100 K, 300 K, 600 K, 1000 K) at pressure $P_0$ (200 GPa, 300 GPa, 400 GPa, 500 GPa, 600 GPa, 700 GPa), (ii) integrate along $T$ between 100 K and 2000 K, and finally (iii) integrate along pressure between 100 GPa and 800 GPa. As shown in Supplementary Fig. 8, these different TI routes provide consistent estimations of the classical chemical potentials based on the MLP.

In addition, we promoted the MLP results to the PBE level by adding $\mu - \mu^{MLP}$ computed using the free-energy perturbation method, which removes the small residual errors in the MLP partly due to its lack of long-range electrostatics[30,32]. This correction term at different pressure and temperature conditions is illustrated in Supplementary Fig. 9. To account for the influence of NQEs on the free energies, we ran path-integral molecular-dynamics (PIMD) simulations on small systems with 64 $H_2O$ formula units using LAMMPS in conjunction with the i-PI code[56]. These PIMD simulations were performed employing the MLP, with 32 beads for all phases considered at the relevant constant pressure and temperature conditions. NQEs on the chemical potentials were taken into account by integrating the quantum centroid virial kinetic energy $\langle E_k \rangle$ with respect to the fictitious 'atomic' mass $\widetilde{m}$ from the classical (i.e., infinite) mass to the physical masses $m$[57–59], $\Delta\mu^{NQE}(P,T) = 2 \int_0^1 (\langle E_k(1/y^2) \rangle / y) \, dy$, where $y = \sqrt{m/\widetilde{m}}$. The contribution from NQEs to the chemical-potential difference is small ($< 5$ meV per molecule). The NQE contributions to the chemical-potential difference as a function of pressure and temperature are illustrated in Supplementary Fig. 9.

### Metadynamics simulations

The CVs (the anisotropy and the tilt of the simulation supercell) were chosen such that the X(orthorhombic), X(cubic), P*bcm*, and P*bcm*-like structures with different stacking sequences were distinguished on the FES. The length of each metadynamics run was $2 \times 10^6$ time steps. For each 400 time steps, an adaptive Gaussian bias[29], which covers the space of 400 time steps in the collective variables, was added. The height of the Gaussian bias decayed with the well-tempered scheme[28] using a bias factor of 100.

### Data availability

All original data generated for the study are in the SI repository https://github.com/BingqingCheng/highP-ice.

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

## Acknowledgements

We thank Chris Pickard for providing the initial structures of high-pressure ice phases and for useful advice. A.R. and B.C. acknowledge resources provided by the Cambridge Tier-2 system operated by the University of Cambridge Research Computing Service funded by EPSRC Tier-2 capital grant EP/P020259/1. M.B. was supported by the European Union within the Marie Skłodowska-Curie actions (xICE grant 894725) and acknowledges computational resources at North-German Supercomputing Alliance (HLRN) facilities. S.H. and M.M. acknowledge support from LDRD 19-ERD-031 and computing support from the Lawrence Livermore National Laboratory (LLNL) Institutional Computing Grand Challenge programme. F.C. acknowledges support from the US DOE Office of Science, Office of Fusion Energy Sciences. Lawrence Livermore National Laboratory is operated by Lawrence Livermore National Security, LLC, for the U.S. Department of Energy, National Nuclear Security Administration under Contract DE-AC52-07NA27344.

## Author contributions

B.C. conceived the research; all authors contributed to the design of the research; A.R., M.B., S.H. and B.C. performed the research; A.R., M.B., F.C., M.M. and B.C. wrote the paper; all authors commented and helped improve the paper.

## Competing interests

The authors declare no competing interests.
