## [Peer Review File · Nature Communications]

REVIEWER COMMENTS

Reviewer #1 (Remarks to the Author):

Water/ice has a very complicated phase diagram at high pressures and high temperatures. In this work, Reinhardt et al applied DFT and machine learning MD methods to study the phase transitions between high pressure ices: ice VII, VII', VII'', and X. The efficient machine learning force fields enable longer and larger atomic simulations, which are important for converged thermodynamic quantities. However, it seems the main results have been reported in the previous studies (e.g. refs 1-16), so I am not sure the originality. There are also many problems about the simulation methods and results, for which I have the following comments:

(1) It was reported that there is a second-order phase transition between ice VII' and X (e.g., ref 7), but the authors said "ices VII, VII' and X are the same thermodynamic phase under different conditions". Why? Can the authors' simulations exclude the second-order phase transition? How to explain the nonsymmetric O–H··O bonds become symmetric if there is no phase transition?

(2) What are the statistic uncertainties in the simulations and free energy calculations? How long are the MD simulations? The authors should show them, and explain if they are small enough to identify phase transitions.

(3) "The contribution from NQEs to the chemical-potential difference is small (< 5 meV per molecule)." The NQE contribution strongly depends on temperature. I suggest the authors to give the temperature dependent contribution from NQEs in the SI.

(4) The machine learning potential (MLP) is trained on different homogenous ice phases using first principles data, it should be verified the MLP is accurate enough to study the phase transitions which involve inhomogeneous interfaces and boundaries. Additionally, the long-range interactions are often very important for the solid phase; however, the MLP used here has only short-range interactions. Is it good enough to predict the phase transitions of ices, considering that the entropy energies of various ice phases differ only little at certain pressures? For example, Fig. S1 shows that revPBE0-D3, PBE, and MLP have different transition pressures between ice X and Pbcm.

(5) The Gibbs–Duhem relation is normally used to describe the chemical potential difference of two states in the coexisting phase. The authors should prove its applicability when using the relation to study the transition of two single phases.

(6) In Fig. 3d, What is the structure SF1? It seems not at the free energy local minimum. It also seems that the free energy contour has different local minimum for the same crystal structure; it may be due to the inappropriate CVs the authors used, which cannot well consider the symmetry of crystals. The inappropriate CVs may give unreliable results. Is it good to try some order parameters for water/ice as CVs?

(7) The manuscript lacks the methodology details, e.g., the details of training the MLP and calculating the thermodynamic integration.

(8) For the X–Pbcm transition, the simulation cell contains 64 water molecules. Is it big enough to capture the solid-solid transition pathway?

(9) The caption for Fig. 2(c) is very confusing. Why do the cell parameters increase with pressure above ~300 GPa for ice X and Pbcm? Does it mean that the volume increases with pressure?

Reviewer #2 (Remarks to the Author):

The manuscript by Reinhardt et al. investigates the nature of the phase transitions among several high-pressure phases of ice at the boundary between molecular and superionic. The authors use a combination of molecular dynamics and thermodynamics approaches to resolve

whether transitions among ice VII, VII', VII'' and X are first-order or not. These simulations identify a coexistence line between the ice VII' and VII'' phases and provide convincing evidence that there is a first-order transition between these two phases, whereas the other "phases" appear to be different manifestations of the same thermodynamic phase, as no discontinuities in their thermodynamic properties are observed. Additionally, the authors characterize the phase transitions of ice X, which exhibits an unusual behavior beyond 300 GPa, as it converts into a Pbcm phase rapidly, but without a jump in enthalpy. The latter Pbcm phase has never been observed experimentally and the authors speculate about the possibility of detecting it in future experiments.

The manuscript is compelling and the introduction is very well-written, as it illustrates the intricate and fascinating phase behavior of high-pressure ice. The authors master a wealth of simulation techniques and combine them appropriately to address the thermodynamics of high-pressure ice. Sufficient details are provided for most of the simulations and data reproducibility would be fully enabled once a repository is made available. The discussion of the results is interesting the claims are supported by data. For these reasons, I would be happy to see this manuscript published in Nature Communications.

Before publication, I would recommend that the following minor issues are addressed:

- 1 - provide a reference to the well-tempered metadynamics method and the details of how these simulations have been carried out
- 2 - As CV1 is a measure of anisotropy, it may be worth normalizing CV1 in Figure 4 b-e so that the results at different pressures are directly comparable. It would be nice if it were indicated where cubic ice X would be located on the diagram when it becomes a saddle point (beyond 300 Gpa).
- 3- Do the authors have any idea as to why the cubic symmetry of ice X breaks at a certain pressure?
- 4- On page 4 it is written that "the integral [in TI] will diverge" for a first-order transition. In fact, a numerical integral won't possibly diverge, but the thermodynamic loop would have hysteresis.
- 5- I do not think that the (c) and (d) panels in figure 1 are particularly useful to the discussion of the results in this form. I would recommend either moving them to SI or extending the discussion/description of these figures.
- 6- Is there any experimental evidence of distinct VII'(T) and VII'(R) phases?
- 7- The authors use paragraph headers in the section on the X-Pbcm Transition. I normally find that paragraph headers (or subsections) enhance the clarity of an article, but in this case, they are used a bit inconsistently: I would recommend that either they are removed or they are used along the whole results section of the manuscript.
- 8- 13 lines to the end of page 2 "larger [...] that" should be "larger than".

Reviewer #3 (Remarks to the Author):

I really enjoy reading Cheng's papers so I was very pleased to review this paper. Firstly, the positive aspects of the work: 1) it's a superbly executed study displaying state of the art simulation know-how. Outstanding work - keep it up. 2) it's a very well written paper that draws the reader in and communicates detail whilst providing perspective - very well balanced. 3) the findings are a little surprising to me. I expected small but non-negligible barriers.

Now to my issues with this work 1) the title - it's not especially clear and certainly not captivating. 2) the findings are interesting in and of themselves and I had not made the structural connection between ice VII and X but I am not sure how this helps the community in a profound way. How can the result be exploited either conceptually or practically? Again, it's not an obvious result but I don't immediately see the impact it could have beyond what has already been found. 3) it feels more suitable for a specialist physics journal like PRL, a feeling that is exacerbated by the formatting.

Are the results noteworthy? yes.

Will the work be of significance to the field and related fields? My feeling is that the impact of this work may not be so high (in comparison to Cheng's other work) and I am unclear on how other

fields may view its significance.

Does the work support the conclusions and claims, or is additional evidence needed? yes

Are there any flaws in the data analysis, interpretation and conclusions? Do these prohibit publication or require revision? No, although it appears only one realisation of the ice VII network has been used and ideally the findings would be demonstrated for another arrangement of protons (although I do not expect any change in the key results).

Is the methodology sound? Does the work meet the expected standards in your field? yes

Is there enough detail provided in the methods for the work to be reproduced? Probably.

So overall, I am, regrettably, not convinced that this work belongs in Nat. Comm.

We thank the referees for their careful reading of our manuscript. We have made a number of changes to the manuscript in response to their comments, and have highlighted these in blue in the revised version of the text. In what follows, we respond to each of the points raised by the referees.

Reviewer #1 (Remarks to the Author):

Water/ice has a very complicated phase diagram at high pressures and high temperatures. In this work, Reinhardt et al applied DFT and machine learning MD methods to study the phase transitions between high pressure ices: ice VII, VII', VII'', and X. The efficient machine learning force fields enable longer and larger atomic simulations, which are important for converged thermodynamic quantities. However, it seems the main results have been reported in the previous studies (e.g. refs 1-16), so I am not sure the originality. There are also many problems about the simulation methods and results, for which I have the following comments:

We would like to thank the referee for reviewing our manuscript and giving constructive suggestions. We would like to stress that although the high-pressure ice phases are a well-studied topic, none of the previous studies [Refs 1-16] probes the thermodynamic phase behaviour of the ices. More specifically, it is not clear which of these ice phases are actually different phases of matter separated by first-order phase boundaries. It is also unknown how one phase transforms to another at finite temperatures. To highlight these points, we have updated the title to emphasise the “thermodynamics” aspect of our contribution.

(1) It was reported that there is a second-order phase transition between ice VII' and X (e.g., ref 7), but the authors said "ices VII, VII' and X are the same thermodynamic phase under different conditions". Why? Can the authors' simulations exclude the second-order phase transition? How to explain the nonsymmetric O—H···O bonds become symmetric if there is no phase transition?

Ref. 7 does not seem to provide any direct evidence that there is a second-order phase transition between ices VII' and X; the manuscript shows that there is no discontinuity in the energy or the elastic constants between ices X and VII', and we presume they mean ‘second-order’ in a loose sense as synonymous with a ‘continuous’ phase transition. Of course it is true that there is a structural change – in the sense that there is a symmetry in ice X that is absent in ice VII' – but this does not seem to translate into a thermodynamic signature of a phase transition. A phase transition is of second order in the Ehrenfest classification if the second derivatives of the free energy with respect to pressure or temperature are discontinuous. In our case, we find no evidence for such a discontinuity, as evidenced by the fact that the first derivatives, which correspond to the density and the enthalpy, appear to be not only continuous, but without any sudden change of gradient across the X–VII' transition. We show these curves in what is now Fig. S1 [in the initial submission this was Fig. 1(c,d)]. It may be that there are some subcritical conditions where a thermodynamic phase transition exists between the two phases, but it does not appear to be in the pressure–temperature range we have investigated. We also cannot of course exclude the possibility that some higher derivative of the free energy

is discontinuous, but on the basis of the data we have, this appears to be unlikely, and this is why we have said that these are the 'same' thermodynamic phase.

In order to reduce the scope for confusion, we now explicitly mention the smoothness of the density and enthalpy curves and clarify that ice X and ice VII' are structurally distinct, but there is no apparent thermodynamic phase transition between them.

(2) What are the statistic uncertainties in the simulations and free energy calculations? How long are the MD simulations? The authors should show them, and explain if they are small enough to identify phase transitions.

Typical MD simulation lengths are of about 20 to 60 ps at each set of conditions considered. Since we generally run simulations along isotherms and isobars, this means that equilibration is usually facile, since the pressure and temperature change only by a small amount for each data point. We start with a pre-equilibrated structure at the conditions of interest. For isotherms, we then gradually change the pressure in steps of between 0.3 GPa and 4 GPa, depending on how close to the limits of metastability our system is. For isobars, we change the temperature in steps of between 5 K and 60 K, again depending on the proximity to the limits of metastability. At each step, we first allow a brief equilibration of 2 ps, followed by a data collection run of ~30 ps. For data collection, we normally take samples every 100 timesteps.

In these runs, we find averages of the appropriate integrand for the thermodynamic integration. These have a natural thermal fluctuation; for example, the standard deviation of the solid-phase density at 1000 K and 60 GPa [Fig. S1(a)(ii); i.e. what was Fig. 1c in the initial submission] is 0.006 g cm^{-3} , while the standard error of the mean is $0.00023 \text{ g cm}^{-3}$, which falls within the line thickness. The corresponding values for the enthalpy [Fig. S1(b)(ii)] are 0.05 eV and 0.0019 eV, respectively, again within the line thickness. Moreover, since we have many data points along each isotherm and isobar, the error due to thermal fluctuations is negligible when determining a polynomial fit across many points.

In free-energy calculations, as discussed by Vega and co-workers [ref. 23], it is difficult to determine the errors in chemical potentials systematically, and *a posteriori* consistency checks are usually undertaken instead, such as the ones we have described in the manuscript (e.g. computing coexistence using both chemical potential calculations and direct-coexistence calculations, and integrating along different isotherms and isobars to the same point). To determine an estimate of the errors in the chemical potentials, we have computed the mean prediction bands at a 95% confidence level for polynomial fits to the integrands along isobars and isotherms. Propagating this error into chemical-potential calculations results in an estimated error that is often again smaller than the line widths (e.g. in Fig. 1d), but is larger whenever any extrapolations are required (e.g. in Fig. 1c). We have added lines indicating these prediction intervals to Fig. 1, although the only place they can be seen is Fig. 1c, since in all other cases the line widths are larger than this interval. Of course this kind of error propagation cannot account for any systematic errors; these can be mitigated by the consistency checks mentioned above.

We have now added more details about the set-up of the MD simulations and the numerical analysis of errors into the Methods section of the manuscript (page 8), and we have added mean prediction bands to Fig. 1c.

(3) "The contribution from NQEs to the chemical-potential difference is small (< 5 meV per molecule)." The NQE contribution strongly depends on temperature. I suggest the authors to give the temperature dependent contribution from NQEs in the SI.

The contribution from NQEs to the chemical-potential difference is available in the Fig. S8(c) of the SI, and we expanded a sentence in the Methods section to indicate this.

(4) The machine learning potential (MLP) is trained on different homogenous ice phases using first principles data, it should be verified the MLP is accurate enough to study the phase transitions which involve inhomogeneous interfaces and boundaries. Additionally, the long-range interactions are often very important for the solid phase; however, the MLP used here has only short-range interactions. Is it good enough to predict the phase transitions of ices, considering that the entropy energies of various ice phases differ only little at certain pressures? For example, Fig. S1 shows that revPBE0-D3, PBE, and MLP have different transition pressures between ice X and Pbcm.

The referee is right that the MLP has only short-range interactions, which may be problematic for describing inhomogeneous interfaces and boundaries. We have compensated for this shortcoming of the state-of-the-art MLP in two ways:

1. Interfaces are irrelevant in our free-energy estimations using the MLP, since we always use pure, bulk (ice or liquid) phases when integrating over isotherms or isobars. We have however benchmarked these chemical-potential calculations with direct-coexistence simulations, in which there is an explicit interface. Since the two approaches give coexistence points which agree very well, this suggests that the interface does not dominate the thermodynamic behaviour even in direct-coexistence simulations given the large system sizes we have used for these calculations. Finally, when computing free-energy barriers in the context of the Pbcm–X transition, we note that this transition occurs via the shuffling of the {100} planes, which does not involve any nucleation or other interfacial phenomena.
2. For computing the chemical-potential difference between the X and the Pbcm phases, we promoted the MLP results to the PBE level by adding $\mu - \mu^{MLP}$ computed using the free-energy perturbation method, which removes the small residual errors in the MLP partly due to its lack of long-range. This correction term at different pressure and temperature conditions is illustrated in Fig. S8. After the correction, the chemical-potential difference is at the PBE level, without any assumptions from the MLP.

To address the referee's concerns, we have now emphasised these points in the manuscript by expanding the description of the MLP->DFT correction procedure in the Methods part of the

paper (page 9), and by explaining that the shuffling transition does not entail nucleation on p. 5 of the manuscript.

(5) The Gibbs–Duhem relation is normally used to describe the chemical potential difference of two states in the coexisting phase. The authors should prove its applicability when using the relation to study the transition of two single phases.

We are slightly unsure what the referee means by this. The Gibbs–Duhem relation comes directly from the fundamental equation of chemical thermodynamics for a one-component system, i.e. $dG = V dP - S dT + \mu dN$, and the Euler relation $G = N\mu$, which gives $dG = N d\mu + \mu dN$. Comparing the two results gives the Gibbs–Duhem relation, $N d\mu = V dP - S dT$, which can then be integrated in pressure at constant temperature. We use this relation to determine the chemical potential of a single phase as a function of pressure, i.e. the procedure does not depend on there being two phases. Perhaps some confusion has arisen because the name ‘Gibbs–Duhem integration’ is sometimes used for an unrelated procedure of numerically integrating the Clapeyron equation ($\partial T/\partial P = \Delta V/\Delta S$) along a coexistence curve (Kofke, *Mol. Phys.* **78**, 1331 (1993)), which relies on an initial determination of the coexistence point of two phases. We have not used this approach, as small errors can rapidly result in the calculated coexistence curve deviating from the true coexistence curve. To reduce the scope for misunderstanding, we have now written out the Gibbs–Duhem relation explicitly in the manuscript.

(6) In Fig. 3d, What is the structure SF1? It seems not at the free energy local minimum. It also seems that the free energy contour has different local minimum for the same crystal structure; it may be due to the inappropriate CVs the authors used, which cannot well consider the symmetry of crystals. The inappropriate CVs may give unreliable results. Is it good to try some order parameters for water/ice as CVs?

We thank the referee for pointing this out; indeed we did not describe clearly what are the SF1 and the SF2 structures in the original submission. These two structures are similar to Pbcm, but with different stacking sequences, and they have excess 0 K enthalpies compared to the Pbcm phase (see the plot below). These two structures are analogous to the P2₁2₁2 and P4₂2₁2 phases illustrated in Fig. 2a. We have added these explanations in the main text.

The plot below shows the 0 K enthalpy for different structures. The wiggly parts for the curves of SF1 and SF2 come from the difference in the proton positions in the relaxed structures.

The Pbcm structure has two local minima on the FES, but this is not because of the choice of the CV, but rather because the Pbcm structure can fit into the simulation box in two different ways. This is not related to the choice of the CV, but simply stems from the specific crystal structure of the Pbcm phase. We have updated the illustrations of the atomic snapshots in Fig. 3d, and added additional sentences in the main text, in order to explain this.

We hope the above explanation about the two minima for Pbcm already resolves the concerns from the Referee regarding the choice of the CV. The current choice of the CVs (the anisotropy and the tilt of the simulation cell) are efficient in sampling X, Pbcm and other structures with different stacking fault sequences. To the best of our knowledge, no other well-established CVs can be used for this purpose: no existing CVs directly distinguish the space groups of crystals at finite temperatures. Steinhardt bond order parameters and their locally-averaged versions are good at distinguishing liquids, fcc and bcc solids, but are not sensitive to the Pbcm-like structures with different stacking sequences.

(7) The manuscript lacks the methodology details, e.g., the details of training the MLP and calculating the thermodynamic integration.

We have now included more details of the thermodynamic integration, in particular how we compute averages in MD simulations (at the end of the first paragraph of the ‘MLP MD simulation details’ part of the Methods section on p. 8) that are then used in thermodynamic integration (i.e. the equations of the second paragraph). We have also expanded on the error analysis (in response to point (2) above, as the last paragraph of this part of the Methods section, also on p. 8). Finally, we have now included the specific integral used in the thermodynamic integration taking into effect nuclear quantum effects (in the last paragraph of the Methods section, p. 9).

The details for training the MLP are provided in our previous paper [Cheng 2021, Ref 18]. In particular, the training set for the MLP is publicly available at <https://github.com/BingqingCheng/superionic-water>

(8) For the X–Pbcm transition, the simulation cell contains 64 water molecules. Is it big enough to capture the solid-solid transition pathway?

The transition pathway from X to Pbcm involves a highly collective shuffling of the {100} planes in X, which is a diffusionless, martensitic transition. Such a transition pathway, unlike nucleation and growth, is insensitive to system size. To confirm this, we performed metadynamics simulations using 8 times larger system sizes (512 molecules). Three snapshots of atomic positions are provided below. Essentially, the transition mechanism (the shuffling of the planes and the possible formation of structures with different stacking sequences) is identical in the smaller and the larger systems. We have added these results from the large system to the SI (Fig. S6). However, the computation of the free-energy surfaces of larger systems is much harder to converge, so we present the results from smaller systems in the manuscript.

Structures obtained from metadynamics simulations using 512 water molecules.

(Left: ice X structure. Middle: structure with stacking faults, similar to SF2 in Fig. 3d. Right: also a structure with stacking faults, similar to SF1 in Fig. 3d.)

(9) The caption for Fig. 2(c) is very confusing. Why do the cell parameters increase with pressure above ~300 GPa for ice X and Pbcm? Does it mean that the volume increases with pressure?

We are sorry for the confusion. In fact in Fig. 2c we show all three cell parameters (l_{xx} , l_{yy} and l_{zz}). Although some cell parameters for the X(orthorhombic) and Pbcm increase with pressure, the other cell parameters decrease even faster with pressure, so the volume always decreases with pressure, as it should. We show the molar volume as a function of pressure explicitly in Fig. S3c in the SI. We have updated the legend of Fig. 2c (see below) and added explanations in the figure caption to try to minimise any confusion.

Reviewer #2 (Remarks to the Author):

The manuscript by Reinhardt et al. investigates the nature of the phase transitions among several high-pressure phases of ice at the boundary between molecular and superionic. The authors use a combination of molecular dynamics and thermodynamics approaches to resolve whether transitions among ice VII, VII', VII'' and X are first-order or not. These simulations identify a coexistence line between the ice VII' and VII'' phases and provide convincing evidence that there is a first-order transition between these two phases, whereas the other "phases" appear to be different manifestations of the same thermodynamic phase, as no discontinuities in their thermodynamic properties are observed. Additionally, the authors characterize the phase transitions of ice X, which exhibits an unusual behavior beyond 300 GPa, as it converts into a Pbcm phase rapidly, but without a jump in enthalpy. The latter Pbcm phase has never been observed experimentally and the authors speculate about the possibility of detecting it in future experiments.

The manuscript is compelling and the introduction is very well-written, as it illustrates the intricate and fascinating phase behavior of high-pressure ice. The authors master a wealth of simulation techniques and combine them appropriately to address the thermodynamics of high-pressure ice. Sufficient details are provided for most of the simulations and data reproducibility would be fully enabled once a repository is made available. The discussion of the results is interesting the claims are supported by data. For these reasons, I would be happy to see this manuscript published in Nature Communications.

We thank the referee for their reviewing of our manuscript and for their positive assessment.

Before publication, I would recommend that the following minor issues are addressed:

1 - provide a reference to the well-tempered metadynamics method and the details of how these simulations have been carried out

We have added references to the well-tempered metadynamics [Barducci 2008] and the paper that introduced adaptive bias [Branduardi 2012] (at the start of page 5).

In the Methods section (on page 9), we added a paragraph on the details of the simulations:

“The CVs (the anisotropy and the tilt of the simulation supercell) are chosen such that the X(orthorhombic), X(cubic), Pbcm and Pbcm-like structures with different stacking sequences are distinguished on the FES. The length of each metadynamics run was 2×10^6 time steps. For each 400 time steps, an adaptive Gaussian bias [Branduardi 2012], which covers the space of 400 time steps in the collective variables, is added. The height of the Gaussian bias decays with the well-tempered scheme [Barducci 2008] using a bias factor of 100.”

2 - As CV1 is a measure of anisotropy, it may be worth normalizing CV1 in Figure 4 b-e so that the results at different pressures are directly comparable. It would be nice if it were indicated where cubic ice X would be located on the diagram when it becomes a saddle point (beyond 300 GPa).

We thank the referee for the suggestions.

For the normalisation of the CVs, it is not obvious which reference structure to use. This is because several structures are explored during the metadynamics simulations, including X, Pbcm and structures with stacking faults. If we use the average shape of either of these structures to normalise the CVs, this can cause confusion at some pressures, e.g. Pbcm is not stable at low pressures and collapses into ice X, ice X has different degrees of anisotropy at different pressures.

For the location of cubic ice X on the free-energy surfaces, we think it is a great idea, and we thank the referee for the suggestion. In the updated version of Fig. 3, we used green diamond symbols to indicate the location of X(cubic) in the CV space. It can be seen that at 200 GPa, the cubic X phase is at a minimum on the free-energy surface, and at higher pressures it becomes a saddle point.

3- Do the authors have any idea as to why the cubic symmetry of ice X breaks at a certain pressure?

The breaking of the cubic symmetry in ice X is indeed very interesting. We added the following discussion in the manuscript on p. 6:

“(the X(cubic) structure is at a saddle point of the potential-energy surface of the system above ~400 GPa, as suggested by its imaginary vibrational modes.) By contrast, X(orthorhombic) has no imaginary modes in all our phonon calculations performed at pressures between 100 GPa and 600 GPa.”

– and –

“The breaking of the cubic symmetry of ice X probably arises from the dynamic instability revealed by the imaginary phonons, which was previously reported in Ref. [Caracas2008]. A similar breaking of cubic symmetry has also been observed for other systems [Militzer2010, Raty2020]. Although Ref.~[Caracas2008] suggests that the instability leads to the X--Pbcm

transition, here we suggest that the cubic–orthorhombic transition is an alternative way of gaining dynamical stability.”

4- On page 4 it is written that "the integral [in TI] will diverge" for a first-order transition. In fact, a numerical integral won't possibly diverge, but the thermodynamic loop would have hysteresis.

Thank you; indeed the wording was clumsy, as when integrating over a discontinuity numerically, this may not be apparent at all, but the integral will have different values (i.e. hysteresis) across a thermodynamic loop. We have tightened the wording now (p. 3 of the manuscript).

5- I do not think that the (c) and (d) panels in figure 1 are particularly useful to the discussion of the results in this form. I would recommend either moving them to SI or extending the discussion/description of these figures.

These panels were meant to illustrate the narrowness of the range of metastability under most conditions of interest and the smoothness of the X/VII' curves, but we agree that they did seem perhaps a little too prominent. As advised, we have now split off panels (c) and (d) into the SI.

6- Is there any experimental evidence of distinct VII'(T) and VII'(R) phases?

Our simulations suggest that these are not distinct thermodynamic phases, but simply the same phase exhibiting different behaviour under different conditions, which makes it more difficult to characterise such 'phases' experimentally. Indeed to our knowledge the difference between the translation-dominated and rotation-dominated diffusion regimes, as identified by Zhuang et al. (Ref. 19 of the manuscript), has not been investigated experimentally. The work of Queyroux et al. (Ref. 8 of the manuscript) comes closest by investigating the difference between different superionic regimes (i.e. in the VII' and VII'' phases), but they note that it is difficult to deduce the positions of protons from XRD experiments due to their weak signal. While we expect the nature of the dynamics in ice VII' under different conditions is likely to be important for mechanistic reasons in phase transitions and perhaps also in terms of conductivity mechanisms, it will no doubt be difficult to probe directly in experiment. Hence computer simulations, where such differences are easier to elucidate, are likely to play an important role in understanding the phase behaviour under these conditions.

7- The authors use paragraph headers in the section on the X-Pbcm Transition. I normally find that paragraph headers (or subsections) enhance the clarity of an article, but in this case, they are used a bit inconsistently: I would recommend that either they are removed or they are used along the whole results section of the manuscript.

We have now removed the paragraph headers, as advised.

8- 13 lines to the end of page 2 "larger [...] that" should be "larger than".

We have corrected this typo; thank you.

Reviewer #3 (Remarks to the Author):

I really enjoy reading Cheng's papers so I was very pleased to review this paper. Firstly, the positive aspects of the work: 1) it's a superbly executed study displaying state of the art simulation know-how. Outstanding work - keep it up. 2) it's a very well written paper that draws the reader in and communicates detail whilst providing perspective - very well balanced. 3) the findings are a little surprising to me. I expected small but non-negligible barriers.

Now to my issues with this work 1) the title - it's not especially clear and certainly not captivating. 2) the findings are interesting in and of themselves and I had not made the structural connection between ice VII and X but I am not sure how this helps the community in a profound way. How can the result be exploited either conceptually or practically? Again, it's not an obvious result but I don't immediately see the impact it could have beyond what has already been found. 3) it feels more suitable for a specialist physics journal like PRL, a feeling that is exacerbated by the formatting.

AUTHORS:

(1) We have changed the title to one which reflects the manuscript contents more clearly and which – we hope – is also captivating. The new title is “Thermodynamics of high-pressure ice phases revealed by atomistic simulations”.

(2) We have added two paragraphs on potential implications to the end of the Discussion section (page 8):

“Experimental investigations on the high-temperature and high-pressure water ices are notoriously difficult, but recent technical breakthroughs could open new opportunities to test the predictions from this work in the future. For example, neutron scattering was used to investigate the atomic structure of ice VII at room temperature up to ~100 GPa and revealed subtle pressure-induced changes of the proton disorder [Guthrie 2019]. Pulsed

internal heating and alternating-current calorimetry in the diamond anvil cell (DAC) is a promising technique for investigating the nature of phase transitions experimentally [Geballe 2021, Geballe 2017]. A combination of resistive heating, near- and mid-infrared laser heating in the DAC to study the structure of dense ice at high temperature with X-ray diffraction has provided evidence for phase transformations consistent with two superionic ices having either a bcc or a fcc oxygen sublattice. CO₂ laser heating in the DAC has also recently been shown to release deviatoric stresses and enable collection of high-quality powder X-ray diffraction and Raman scattering [Grande 2022]. Finally, laser-driven dynamic compression has enabled the investigation of the thermodynamics and atomic structure to several hundred GPa and revealed evidence for superionic ice XVIII [Millot 2018, Millot 2019]. With the rapid evolution and improvement of experimental methods to investigate material properties at extreme conditions, we expect that this work will provide insights into the interpretation of experimental data as well as input for new experiments.”

“The precise nature of phase transformations in dense ices at high temperatures could also affect our understanding of large icy worlds in our solar system such as Ganymede, Titan and Callisto [Journaux 2020] as well as more distant water-rich exoplanets [Noack 2016, Weiss 2016, Lingam 2019, Nixon 2021] as the presence (or absence) of discontinuous phase transitions can induce (or suppress) thermal boundaries which would affect mass and heat flows and have a significant role in shaping the internal structure and evolution of these bodies.”

(3) Although we believe the journal will typeset the manuscript in due course, we have changed the formatting so that it is more similar to that used by Nature Communications rather than PRL.

Are the results noteworthy? yes.

Will the work be of significance to the field and related fields? My feeling is that the impact of this work may not be so high (in comparison to Cheng's other work) and I am unclear on how other fields may view its significance.

Does the work support the conclusions and claims, or is additional evidence needed? yes
Are there any flaws in the data analysis, interpretation and conclusions? Do these prohibit publication or require revision? No, although it appears only one realisation of the ice VII network has been used and ideally the findings would be demonstrated for another arrangement of protons (although I do not expect any change in the key results).

We have in fact used more than one ice VII network to test for both finite-size effects and the influence of proton disorder. We have not found any measurable difference in these different simulations. We have now made this point explicitly in the manuscript.

Is the methodology sound? Does the work meet the expected standards in your field? yes
Is there enough detail provided in the methods for the work to be reproduced? Probably.

So overall, I am, regrettably, not convinced that this work belongs in Nat. Comm.

REVIEWER COMMENTS

Reviewer #1 (Remarks to the Author):

I would like to thank the authors for answering our questions.

Although the authors reported some interesting findings about the thermodynamics of high-pressure ices, I am still not very sure about the significance, considering that the MLP was trained in their previous works [Cheng 2021, Ref 18], and the ice phases was discovered. And the parts of superionic ice were presented in [Cheng 2021, Ref 18]. I also feel the random stacking and tilt phase transition between the Pbcm and x ice are not very novel, which is common in crystal structures.

"In our case, we find no evidence for such a discontinuity, as evidenced by the fact that the first derivatives, which correspond to the density and the enthalpy, appear to be not only continuous, but without any sudden change of gradient across the X-VII' transition."
The evidence is not persuading. You can not judge the derivative of a curve is continuous by visualizing the curve itself. The MLP approximates the DFT force potential. Considering the error of MLP and statistical uncertainties, it is impossible to rule out that the derivative of density is not continuous. The enthalpy is not the first derivative of the free energy ($G = H - TS$).

I appreciate that the authors added the corrections to the chemical potentials. To validate the MLP, I suggest the authors to conduct the enhanced sampling simulation (e.g., metadynamics) at the DFT level at the same condition as in Fig. 3(d), can the authors observe the same phenomena? I think the size is affordable. The author can at least show the similar convertible phenomena in the DFT metadynamics.

"SF1 and SF2 are structures that are similar to Pbcm but with different stacking sequences. "
Fig 3(d) clearly shows that SF1 is not at a local minimum, but the authors claimed that "the FES has five distinct minima". If the error of free energies is so small as the authors said, we should be able to see the distinct SF1 state easily, but we can not.

In Fig. S4, some words overlap with curves.

We thank the referees for their assessment of our revision. We have made a number of changes to the manuscript in response to the comments, and have highlighted these in blue in the revised version of the text. In particular, we have performed new calculations to address the concern of Reviewer 1. In what follows, we respond to their individual points.

Reviewer #1 (Remarks to the Author):

I would like to thank the authors for answering our questions.

Although the authors reported some interesting findings about the thermodynamics of high-pressure ices, I am still not very sure about the significance, considering that the MLP was trained in their previous works [Cheng 2021, Ref 18], and the ice phases was discovered. And the parts of superionic ice were presented in [Cheng 2021, Ref 18]. I also feel the random stacking and tilt phase transition between the *Pbcm* and *x* ice are not very novel, which is common in crystal structures.

We would like to thank the referee for reviewing our revised manuscript. Indeed the MLP was constructed in our previous work focusing on the superionic phases. We also agree that the diffusionless mechanism of the *Pbcm*-*X* transition is similar to the martensitic transitions in many metals. However, the diffusionless mechanism is only a minor part of our study, and the main findings include the thermodynamic phase behaviours of the various bcc ice phases and the *Pbcm* phase, and the free-energy landscape of the *Pbcm*-*X* transition (including the peculiar change from a barrierless transition to a higher transition barrier at increasing pressures). As detailed in the conclusion part of the paper, these findings will help guide and interpret high-pressure experiments, as well as have implications on the structures of Ice Giants. Moreover, our methodology combines MLP, enhanced sampling, free-energy methods and path-integral molecular dynamics: compared with our previous study [Cheng 2021], we are therefore able to probe dynamic solid-solid transition processes, which is a big step forward towards the ab initio modelling of material properties beyond equilibrium thermodynamics.

"In our case, we find no evidence for such a discontinuity, as evidenced by the fact that the first derivatives, which correspond to the density and the enthalpy, appear to be not only continuous, but without any sudden change of gradient across the X-VII' transition."

The evidence is not persuading. You can not judge the derivative of a curve is continuous by visualizing the curve itself. The MLP approximates the DFT force potential. Considering the error of MLP and statistical uncertainties, it is impossible to rule out that the derivative of density is not continuous. The enthalpy is not the first derivative of the free energy ($G = H - TS$).

The enthalpy is closely related to a first derivative in the free energy, and a discontinuity in the enthalpy is a characteristic of a first-order phase transition (i.e. a latent heat). This arises directly from the fundamental equation for the Gibbs energy, $dG = V dP - S dT + \mu dN$, from which we have $(\partial G/\partial T)_{P,N} = -S$. Therefore for two phases, α and β , the changeover in gradients at a phase transition is $(\partial G^\alpha/\partial T)_{P,N} - (\partial G^\beta/\partial T)_{P,N} = -\Delta_{\text{trs}}S = -\Delta_{\text{trs}}H / T_{\text{trs}}$, with the last equality following on from the condition that at equilibrium the chemical potentials of the two phases are equal and

so $\Delta_{\text{trs}}G = 0$. Since the temperature changes continuously, the corresponding thermodynamic variable to consider in the context of determining the (Ehrenfest) order of the phase transition is therefore the enthalpy, and this is why we are referencing the enthalpy in our discussion of the order of the phase transition in the manuscript.

It is of course true to say that the underlying phase transition *may* be of second order and the discontinuity in the gradient is very slight, or only arises at the DFT level. However, we are not making a claim that this is definitely not a second-order phase transition, but only that there is no evidence from simulations that would indicate that the phase transition is of second order. We used the word ‘suggesting’ in our previous iteration to intimate this point. However, in the light of the referee’s concerns, we have now emphasised the point that it is ‘on the basis of the MLP phase behaviour’ that ‘there is no evidence that this is even a second-order phase transition’. We have also clarified what we mean by a ‘continuous’ gradient (i.e. not merely visualising the curve, but computing the gradient with a finite-difference method).

I appreciate that the authors added the corrections to the chemical potentials. To validate the MLP, I suggest the authors to conduct the enhanced sampling simulation (e.g., metadynamics) at the DFT level at the same condition as in Fig. 3(d), can the authors observe the same phenomena? I think the size is affordable. The author can at least show the similar convertible phenomena in the DFT metadynamics.

We thank the referee for this helpful suggestion. Such a computation would indeed be meaningful for the further validation of the MLP; however, unfortunately DFT metadynamics simulations are not computationally affordable. Although the system size is sufficiently small to tackle at the DFT level, the required simulation time (nanoseconds) is not. Indeed, this is the very reason why we use the MLP in the first place. We have now added a sentence in the manuscript to highlight this timescale, explaining that metadynamics simulations are only tractable thanks to the use of an MLP.

However, we have performed two sets of additional calculations that are similar in spirit to the DFT metadynamics that the referee suggested and that serve to provide additional justification for the validity of the MLP approach. In particular:

1. We have recomputed the DFT energies for the MLP metadynamics trajectory generated at 400 GPa and 1000 K. A total of 2,500 snapshots from the trajectory were considered, and these include configurations belonging to all five regions of the FES in Fig. 3d. The DFT energies and the MLP energies agree very well, as shown in the two figures below.

(Upper: the potential energies for the 2,500 snapshots predicted by PBE DFT and by the MLP, relative to the mean energy across all the frames. Lower: the parity plot of the PBE and the MLP potential energies for the metadynamics snapshots.)

- To probe the dynamic solid–solid transition mechanism at the PBE DFT level, we performed AIMD *NPT* simulations that lasted 12.5 picoseconds at 1000 K and 400 GPa. All cell parameters were allowed to fluctuate. The starting configuration was cubic X. We observed a facile and diffusionless transition to a distorted *Pbcm* structure. The distortion happens because the simulation cell is not commensurate with the perfect *Pbcm* crystal. We then performed analogous *NPT* simulations using the MLP also at 1000 K, 400 GPa and starting from cubic X, and we observed the exact same transition. We provide snapshots from both sets of the simulations.

(Snapshots from PBE *NPT* MD. Left: $t = 0.01$ ps; right: $t = 1$ ps)

(Snapshots from MLP *NPT* MD. Left: $t = 0.01$ ps; right: $t = 1$ ps)

To summarise, we have performed additional simulations to show that the MLP not only provides faithful descriptions of the relevant structures in metadynamics simulations, but also describes the dynamic transition process accurately.

"SF1 and SF2 are structures that are similar to Pbcm but with different stacking sequences. " Fig 3(d) clearly shows that SF1 is not at a local minimum, but the authors claimed that "the FES has five distinct minima". If the error of free energies is so small as the authors said, we should be able to see the distinct SF1 state easily, but we can not.

The referee is right that from Fig. 3(d), it is difficult to discern whether SF1 is at a saddle point or a shallow local minimum. However, in the FES at other pressures (e.g. Fig. 3(e)), SF1 is at a local minimum of the PES, and what we tried to get across is that these five regions are the relevant states to consider in the range of temperatures and pressures studied. What we are trying to convey is that there are these five structures that have low free energies and are well-sampled in the metadynamics simulations. Our wording was a little careless, and we have now tightened it up by describing these as relevant 'regions' in the FES, rather than 'minima'.

In Fig. S4, some words overlap with curves.

We have edited Fig. S4 and resolved the overlap.

REVIEWER COMMENTS

Reviewer #1 (Remarks to the Author):

I would like to thank the authors for answering my questions. Please see below.

(1) Second order phase transition or not?

In their rebuttal letter, they admitted that "It is of course true to say that the underlying phase transition may be of second order and the discontinuity in the gradient is very slight, or only arises at the DFT level. However, we are not making a claim that this is definitely not a second-order phase transition, but only that there is no evidence from simulations that would indicate that the phase transition is of second order." In the abstract, however, the authors claimed that "Our atomistic simulations show that, amongst these bcc ice phases, ices VII, VII' and X are the same thermodynamic phase under different conditions."

The authors admitted that they cannot exclude the possibility of the second order phase transition, so they can not simply claim that ices X and VII' are "the same thermodynamic phase"; their simulations are not capable of distinguishing the phase boundaries between ice VII' and X, which is very likely due to the statistical nature of machine learning. "The finite difference method" is only a numerical method to approximate derivatives, but definitely can not be used to judge the continuity of derivatives!

Let me give one example: my simulation has such a large numerical uncertainty that it cannot distinguish ice and water using the free energy, but I would never say that my simulation shows that ice and water "are the same thermodynamic phase under different conditions" "within numerical error", or "on the basis of the MLP phase behaviour" "there is no evidence that this is even" a ice-water phase transition.

(2) I am disappointed that " DFT metadynamics simulations are not computationally affordable". "Each set of metadynamics simulations lasts for 3.5 ns". I think it is affordable to have hundreds of picoseconds DFT simulations at the PBE-D3 level or using ADMM for revPBE0-D3. In doing so we will see a rough picture of the X-Pbcm phase transition, which can be used to validate the MLP results at least qualitatively.

The authors said the main findings of this work are "the thermodynamic phase behaviours of the various bcc ice phases and the Pbcm phase, and the free-energy landscape of the Pbcm-X transition.", but my comments suggest that both of them are problematic, so I cannot support publishing it.

Reviewer #1 (Remarks to the Author):

I would like to thank the authors for answering my questions. Please see below.

(1) Second order phase transition or not?

In their rebuttal letter, they admitted that "It is of course true to say that the underlying phase transition may be of second order and the discontinuity in the gradient is very slight, or only arises at the DFT level. However, we are not making a claim that this is definitely not a second-order phase transition, but only that there is no evidence from simulations that would indicate that the phase transition is of second order." In the abstract, however, the authors claimed that "Our atomistic simulations show that, amongst these bcc ice phases, ices VII, VII' and X are the same thermodynamic phase under different conditions."

The authors admitted that they cannot exclude the possibility of the second order phase transition, so they can not simply claim that ices X and VII' are "the same thermodynamic phase"; their simulations are not capable of distinguishing the phase boundaries between ice VII' and X, which is very likely due to the statistical nature of machine learning. "The finite difference method" is only a numerical method to approximate derivatives, but definitely can not be used to judge the continuity of derivatives!

Let me give one example: my simulation has such a large numerical uncertainty that it cannot distinguish ice and water using the free energy, but I would never say that my simulation shows that ice and water "are the same thermodynamic phase under different conditions" "within numerical error", or "on the basis of the MLP phase behaviour" "there is no evidence that this is even" a ice-water phase transition.

Authors:

We think the referee may have misinterpreted what we have written, and is setting up a 'straw man' in their hypothetical ice–water simulation. What we mean when we say 'there is no evidence' is simply that using the usual tools of science, it is impossible to exclude the possibility of something occurring with absolute certainty. Our density and enthalpy data, computed on a very fine grid with small error bars, do not feature *any* signatures that a second-order phase transition separates ices X and VII'. The wording in the abstract reflects this, and is very clear that we are referring to 'our atomistic simulations'.

(2) I am disappointed that " DFT metadynamics simulations are not computationally affordable". "Each set of metadynamics simulations lasts for 3.5 ns". I think it is affordable to have hundreds of picoseconds DFT simulations at the PBE-D3 level or using ADMM for revPBE0-D3. In doing so we will see a rough picture of the X–Pbcm phase transition, which can be used to validate the MLP results at least qualitatively.

Authors:

We have shown in three ways that the MLP can accurately describe the mechanism of the X–Pbcm transition:

- Geometry optimization using revPBE0-D3 and PBE revealed the same barrierless transition that is captured by the geometry optimization employing the MLP.
- In our last revision, we performed PBE AIMD simulations, which show a facile and diffusionless transition from X to a distorted *Pbcm* structure. We also performed analogous simulations using the MLP, and observed the exact same transition.
- In our last revision, we computed the PBE energies for the snapshots collected from a metadynamics trajectory generated using the MLP. The DFT energies and the MLP energies agree very well, meaning that the MLP correctly captures the energetics of the configurations that are relevant for the X–Pbcm transition.

We have thus thoroughly validated the MLP for describing the X-Pbcm transition. The direct AIMD metadynamics simulations suggested by the referee are not useful because even with hundreds of picoseconds of simulations, the FES will not have converged. With such a simulation, one could only hope to get a trajectory with some solid–solid transitions, but such a trajectory with a solid–solid transition has already been obtained from our AIMD simulations.

Furthermore, the computational cost for AIMD metadynamics of hundreds of picoseconds is not affordable: A single point calculation for our system of 64 water molecules takes 40 CPU hours to compute (996 s with 144 CPUs), and a trajectory of 100 ps has 400,000 configurations when using a timestep of 0.25 fs. This means that an AIMD metadynamics trajectory of 100 ps would take 16,000,000 CPU hours to generate. Moreover, the simulation cannot be parallelized in time, but can only be run sequentially, which means that it would take 12.6 years on 144 CPUs to run this 100-ps AIMD simulation. Even if one uses less converged k-point-sampling of just 1 k-point (which is not advisable for this type of high-pressure ice system and may cause artefacts), it will be about 40 times cheaper, but still extremely expensive (115 days using 144 CPUs for a 100-ps simulation). One of the advances of our paper is precisely to be able to run long metadynamics simulations to probe the transition process by exploiting the MLP with first-principles accuracy, which makes AIMD metadynamics unnecessary.

The authors said the main findings of this work are "the thermodynamic phase behaviours of the various bcc ice phases and the *Pbcm* phase, and the free-energy landscape of the *Pbcm*–X transition.", but my comments suggest that both of them are problematic, so I cannot support publishing it.